# The Use of Fibers, Herbal Medicines and Spices in Children with Irritable Bowel Syndrome: A Narrative Review

**DOI:** 10.3390/nu15204351

**Published:** 2023-10-12

**Authors:** Daniela Pop, Radu Samuel Pop, Dorin Farcău

**Affiliations:** 1Third Pediatric Discipline, Mother and Child Department, “Iuliu Hațieganu” University of Medicine and Pharmacy, 400217 Cluj-Napoca, Romania; 2Third Pediatric Department, Clinical Emergency Hospital for Children, 400217 Cluj-Napoca, Romania; 3Nursing Discipline, Mother and Child Department, “Iuliu Hațieganu” University of Medicine and Pharmacy, 400089 Cluj-Napoca, Romania

**Keywords:** irritable bowel syndrome, children, fibers, peppermint oil, herbal remedies

## Abstract

The pathophysiology of irritable bowel syndrome in children involves multiple factors. Thus, treatment options are variable, targeting both diet and the child’s and parents’ behavior via pharmacological and psychological interventions or neuromodulation. Parents are increasingly interested in complementary and alternative therapies for children with irritable bowel syndrome, especially when other treatments have been tried without relieving the child’s symptoms. This paper examines current evidence for the benefits and side effects of herbal remedies and spices in pediatric patients with IBS. The benefits of peppermint oil, STW5, psyllium fiber, Curcuma, ginger, and other herbal medicines are discussed based on findings in the current literature.

## 1. Introduction

Irritable bowel syndrome (IBS) forms part of the functional gastrointestinal diseases, included in the Rome classification, as a “disorder of gut-brain interaction” [1,2]. 

Functional gastrointestinal disorders have a significant impact on pediatric pathology. IBS is reported most frequently among functional abdominal pain diseases, with a pooled prevalence of 8.8%, as concluded by a study conducted by Korterink et al. [3]. The Rome IV classification updated the diagnostic criteria for functional gastrointestinal disorders [1]. After the change in the clinical definition, the prevalence of IBS increased from 2.8% according to Rome III criteria to 5.1% based on the new definition [4].

The etiology of IBS is still unknown. A wide range of mechanisms implicate the microbiota–gut–brain axis, which might be involved in the pathophysiology of IBS. In the top-down model, the initiator [5,6] is the brain, influenced by psychological stress factors, triggering through the autonomic nervous system and the hypothalamic-pituitary-adrenal axis-visceral hypersensitivity, alteration of the gut motility, increased permeability of the mucosa, inflammation, activation of the gut immune system, and changes in the gut microbiota. In the bottom-up model, the process starts with factors influencing the gut (infections, inflammation, distension, microbiota alterations, diet, allergy, immune system), and through the spinal and sympathetic afferents, the brain’s response is altered [6,7]. 

The multitude of underlying causes of IBS leads to a heterogeneity of treatment options that might efficiently relieve the bothersome symptoms that children experience and are worrisome for parents [8]. Consequently, parents seek alternative solutions and often initiate herbal therapies themselves. This initiative often remains undisclosed to the medical professional.

Diagnostic criteria in IBS are based on the description of symptoms, of which the most important is abdominal pain accompanied by altered bowel movements when other structural, inflammatory, or biochemical causes have been excluded [1]. These symptoms recur and persist for at least 2 months. Depending on the bowel habit, there are four subtypes of IBS: predominantly manifested with diarrhea (IBS-D), predominantly associated with constipation (IBS-C), mixed or alternating stool forms (IBS-A), and unclassified (IBS-U) [6].

It is known that dietary fibers promote digestive health by influencing stool consistency and frequency of defecation, the microbiota, and metabolism [9]. Herbal remedies used in traditional medicine have pharmacologically different components, and their synergistic effect could target multiple mechanisms and thus be of benefit in such a complex disease as IBS [10,11]. 

Some reviews have tackled the subject of alternative therapies in children with IBS, either briefly, among other therapies used in this disease [12,13], or in depth, debating the effects of alternative therapies in children with functional abdominal pain disorders [14] or gastrointestinal diseases [15]. This review focuses on the current evidence regarding alternative therapies in children with IBS. We included randomized control trials, cluster-randomized trials, randomized cross-over trials, and prospective observational studies. We also examined systematic reviews published so far.

We searched for articles written in English relevant to this topic using PubMed, Scopus, and Cochrane databases. The keywords that were used were: “irritable bowel syndrome”, “children”, “fiber”, “psyllium”, “peppermint”, “*Mentha piperita*”, “STW5”, “ginger”, “curcuma”, “turmeric”, “artichoke”, “licorice”, “fennel”, “glucomannan”, “guar gum”, “cannabis”, “artichoke”, “herbal remedies”. 

## 2. Dietary Fibers

The term “dietary fibers” refers to carbohydrate polymers and oligomers not hydrolyzed by endogenous enzymes in the small intestine [16]. The fermentation of fibers produces gases and short-chain fatty acids. Short-chain fatty acids stimulate colonic activity by acting on the myenteric neurons and have a role in immunoregulation [17]. An adult’s recommended dietary fiber intake is 25–35 g daily [18]. Children’s recommended daily fiber intake ranges between 10 and 30 g, depending on the patient’s age [19]. There are two main categories of dietary fibers: soluble and insoluble. Insoluble fibers accelerate intestinal transit by bulking the stool and increasing its water content [20]. The main complaints related to dietary fibers are bloating and abdominal pain, mostly related to insoluble fibers [20].

### 2.1. Insoluble Fibers

Corn fibers, lignin, cellulose, hemicellulose, and wheat barn are examples of insoluble fibers. Studies in adult patients with IBS showed no beneficial effects and even exacerbated symptoms, such as constipation, bloating, or abdominal pain [21,22].

In 1985, Feldman et al. [23] published the results of a prospective, double-blind, randomized controlled trial that included children aged 5 to 15 years with recurrent abdominal pain. At that time, functional gastrointestinal disorders in children were not adequately defined and classified, so we do not know how many of the 52 children included in the study had IBS. A total of 26 children received two cookies of 5 g corn fiber each (10 g corn fiber/day) for six weeks. The frequency and intensity of pain attacks decreased in the experimental group (decrease in frequency in 13/26 children-50%), as opposed to the placebo group (decrease in frequency in 7/26 children-27%)-*p* < 0.04 [23]. No significant side effects were described in either group of patients [23].

### 2.2. Soluble Fibers

Psyllium fiber (also named ispaghula) is a moderately water-soluble fiber [24]. It derives from the seeds of Plantago ovata Forsk (Plantago ispaghula Roxb., Fam. Plantaginaceae) [17]. The mucilage extracted from these seeds contains a highly branched polysaccharide, acidic arabinoxylan, a prebiotic that supports beneficial bacteria growth and stimulates short-chain fatty acids production [25]. The gel-forming component of psyllium (arabinoxylan) is partially fermented and retains water in the small intestine [26], increasing stool output and lowering blood cholesterol levels [25]. Psyllium also has an immune modifying effect on gastric epithelial cells [27].

Two randomized controlled trials investigated the benefits of psyllium fiber in children with IBS. A summary of the outcomes of these two studies can be found in Table 1.

In the study of Shulman et al. [28], 37 children with IBS, diagnosed based on Rome III criteria, were included in the study group, and 47 in the placebo group. It was not the expected number of patients. Children included in the study group received psyllium, and those in the placebo group maltodextrin. The frequency of the pain episodes decreased in the group which received psyllium, but the intensity of pain, intestinal microbiome and permeability, breath methane, and hydrogen were similar in the two groups [28].

A recent study from Menon et al. [29] investigated the effects of psyllium on 43 children with IBS compared to the effects of maltodextrin in 38 children with IBS. About one- third of patients, with either IBS-D, IBS-C, or IBS-M, were equally distributed in the two study groups. Psyllium was equally effective in all IBS subtypes. Efficacious pain relief was also observed. The primary data of the study are summarized in Table 1. However, the authors used questionnaires developed for adults, and these might not be appropriate for children, especially for specific age groups. The authors concluded that psyllium is more effective than a placebo in the short term [29].

A randomized placebo-controlled study conducted by Christensen et al. [30] demonstrated no benefits of psyllium in children with recurrent abdominal pain. The study initially included 40 children aged 3 to 15 years, but only 31 completed the study (15 children who received psyllium and 16 children in the placebo group). However, the study included children with recurrent abdominal pain, in which organic causes were excluded, and not specifically children diagnosed with IBS.

Guar gum is a gel-forming galactomannan, a polysaccharide extracted from the seeds of *Cyamopsis tetragonolobus.* Partially hydrolyzed guar gum derives from guar gum and is water soluble, of low viscosity, and non-gelling [31]. It is resistant to digestive enzymes, heat, acid, and salt [32]. Partially hydrolyzed guar gum is highly hydrophilic, accelerates the transit through the digestive tract, and decreases pain by reducing the pressure in the colon. Through metabolization in the colon, partially hydrolyzed guar gum produces short-chain fatty acids modulating microbial growth and acting as a prebiotic [33].

Romano et al. [33] conducted a prospective, single-blind randomized controlled clinical trial in children with IBS who received partially hydrolyzed guar gum in fruit juice, and a placebo group who received only fruit juice. In both groups were included children with IBS-D and IBS-C. Data related to this study are depicted in Table 1. Patients were assessed using the “Birmingham IBS Symptom Questionnaire”, “Wong-Baker Faces Pain Rating Score”, and “Bristol Stool Scale”. The authors concluded that water-soluble fibers may benefit children with IBS, as they may act as a prebiotic, but studies of the long-term effects are needed [33]. 

Glucomannan is a polysaccharide of 1,4-D-glucose and D-mannose, a soluble fiber extracted from the Japanese Konjac plant [34]. Horvath et al. [34] studied the benefits of glucomannan in children aged 7 to 17 years diagnosed with functional gastrointestinal disorders related to abdominal pain, with IBS among them, according to Rome III criteria. It was a double-blind, randomized, placebo-controlled trial. In total, 84 children completed the study (41 children received 2.5 g/day of glucomannan, and 43 children in the placebo group received maltodextrin). The treatment was administered for 4 weeks. The authors aimed to see the proportion of patients who self-reported as having no pain, and that of patients with treatment success. Treatment success was defined as no pain or a decrease of ≥2/6 points on the FACES Pain Scale Revised, the tool used to assess the severity of pain. There was no subgroup analysis, according to a specific diagnosis of functional gastrointestinal disorder, because of the small number of patients in each subgroup. There were no reported adverse events. Four patients in the glucomannan group and two in the placebo group complained of exacerbated symptoms. Overall, 29% of the patients in the glucomannan group declared “no pain” versus 14% in the placebo group. The difference was insignificant (RR = 2.1, 95%CI: 0.87–5.07). Treatment success was found in 23/41 (56%) children in the glucomannan group versus 20/43 (47%) children in the placebo group (RR = 1.21, 95%CI: 0.79–1.83). There were no statistically significant differences between the two groups regarding abdominal cramps or bloating, episodes of nausea and vomiting, changes in loose stools, or constipation [34].

## 3. Herbal Medicinal Preparations and Spices

STW-5 is a herbal combination of nine extracts from medicinal plants (*Iberis amara*, *Angelicae radix*, *Cardui mariae fructus*, *Chelidonii herba*, *Liquiritiae radix*, *Matricariae flos*, *Melissae folium*, *Carvi fructus*, and *Mentha piperita*) [35]. The different components exert an action on the smooth muscles of the gastrointestinal tract; stimulate gastric secretion; have spasmolytic, choleretic, and anti-inflammatory effects; and reduce visceral hypersensitivity [35,36].

There are no randomized placebo-controlled trials to evaluate the benefits and adverse events of STW-5 in children with IBS. We found one prospective observational study which evaluated the therapy with STW-5 in children with functional gastrointestinal disorders, diagnosed based on Rome III criteria [35]. Of the 980 patients aged 3 to 14 years, mean age ± standard deviation (SD) = 7.6 ± 2.9 years, 418 (43%) were diagnosed with IBS. The dose of STW-5 was between 10 and 20 drops, three times a day, depending on the child’s age. A gastrointestinal symptom score validated in adults and developed for patients with functional dyspepsia, and to which a symptom score for lower gastrointestinal disorders was added, was used to assess the clinical data. The patient’s data were re-evaluated after 1 week of treatment. For children with IBS, the gastrointestinal symptoms score decreased from 19.1 ± 8.9 at baseline to 4.8 ± 4.6, with a relative improvement of 75%. The tolerability of STW-5 was rated good or very good in most patients (95%), with only 0.7% of the participants reporting adverse events (skin rash, nausea, vomiting, abdominal pain, and increased gastrointestinal complaints). In addition to being only an observational study, another limit is related to the short duration of the patients’ symptoms. Considering these limitations, the authors concluded that STW-5 improved symptoms of children with functional gastrointestinal disorders, including IBS [35].

Peppermint oil is extracted from *Mentha piperita*, a perennial flowering plant. Its major component is menthol. Peppermint oil is rapidly absorbed. In enteric-coated capsules, its release is delayed and 70% reaches the colon. Menthol blocks calcium channels and reverses acetylcholine, and serotonin-induced smooth muscle contractions. Its primary benefit in IBS seems to be due to this antispasmodic effect. Menthol also acts directly on the enteric nervous system; is a topical analgesic, reducing visceral pain; and has anti-inflammatory, antimicrobial and antifungal effects [37]. 

Kearns et al. [38] performed an exploratory study of the pharmacokinetics of menthol in children with IBS, with a single oral dose of 187 mg. They included in the study six children, aged 7–12 years (10.3 ± 1.9 years), diagnosed with IBS based on the Rome III criteria. Peak plasma concentration was reached after 2.5 to 8 h. The authors concluded, citing similar studies in adults using other peppermint oil-containing products, that the dose-concentration-effect relationship needs further study, and that formulation-specific differences must be considered [38]. 

Kline et al. [39] enrolled 42 children with an IBS diagnosis based on Manning [40] or Rome I criteria in a randomized, double-blind, placebo-controlled study. A 15 item Gastrointestinal Symptom Rating Scale (including symptoms such as abdominal rumbling, belching, abdominal distension, heartburn, urgency for defecation, and stool consistency), a severity symptom scale, and a change of symptom scale were used. Careful neurologic examination, and questions meant to exclude other causes of the symptoms, were also included in the patients’ evaluations. The results of the study are summarized in Table 1. The authors concluded that peppermint oil reduced the severity of pain in a short period of 2 weeks in children with IBS. Long-term effects were not assessed. Other associated symptoms were not influenced [39].

**Table 1 nutrients-15-04351-t001:** Randomized placebo-controlled trials for fibers or herbs used in children with irritable bowel syndrome (IBS).

Reference	Fiber or Herb	Dose Received by the Study Group	Period	Number of Children with IBS	Age	Primary and Secondary Outcomes	Adverse Events	Results
Romano et al. (2013) [33]	Partially hydrolyzed guar gum	5 g/dose in 50 mL fruit juice	4 weeks Follow-up 4 weeks	60 (30 study group/30 control) c-IBS-21 vs. 19 d-IBS 9 vs. 11	Median 12.8 years (range 8–16 years)	reduction in the frequency and intensity of clinical symptoms; correlation with the improvement of character of stool; evaluation of compliance and safety of PHGG in children	-no adverse events	-Treatment success rate-43% vs. 5% (*p* = 0.025); -Significant reduction in the Birmingham IBS score–total score and three subscale scores for diarrhea, constipation, and pain at 4- and 8-week evaluation (median 0 ± 1 vs. 4 ± 1, *p* = 0.025); -Bristol scale improvement (40% vs. 13.3%, *p* = 0.025; -Decrease in the intensity of pain not statistically significant-*p* > 0.05
Shulman et al. (2017) [28]	Psyllium fiber	6 g 7–11 years; 12 g 12 to 18 years	6 weeks	84 (37 children received psyllium and 47 maltodextrin)	7–18 years	-abdominal pain, stool diaries for 2 weeks -Breath hydrogen and methane production -Intestinal permeability -Composition of the microbiome -Psychological factors	-not mentioned	-pain frequency was reduced in the fiber group compared with placebo (8.2 ± 1.2 vs. 4.1 ± 1.3, *p* = 0.03); pain intensity did not differ between the two groups -breath hydrogen and methane production, gastrointestinal permeability similar in the two groups; -stool microbiome did not change in the two groups; -number needed to treat: three.
Menon et al. (2023) [29]	Psyllium fiber	6–12 years→ 6 g/day; 13–18 years→ 12 g/day	4 weeks	81 patients randomized in 2 groups (43 patients in Group A, receiving psyllium and 38 children in Group B receiving placebo-maltodextrin	4–18 years	-patients who achieved complete remission defined as IBS-SSS < 75; -Stool consistency based on Bristol stool chart, severity assessment IBS severity scoring scale (IBS-SSS)-not validated for children, but used in adults (mild, moderate or severe).	-no adverse effects reported by the patients	-remission was attained in 43.9% of children who received psyllium and 9.7% in the maltodextrin group (*p* = 0.0001) -comparable remission rate in all the subgroups (41.7% IBS-D, 50% IBS-C, 41.2%-IBS-M) -number needed to treat: three.
Kline et al. (2001) [39]	Peppermint oil	187 mg peppermint oil/capsule 30–45 kg 3 times 1 capsule >45 kg, 3 × 2 capsules	2 weeks	42 children placebo-arachis oil	8–17 years (mean age 12 years)	-efficacy and clinical usefulness of pH-dependent, enteric coated, peppermint oil capsules	-no side effects reported	-76% of patients receiving peppermint oil versus 19% of patients receiving a placebo reported improvement in the severity of symptom scale; -71% of children in the peppermint group versus 43% in the placebo group had improvements of symptom scale; -no significant changes on the Gastrointestinal Symptom Rating Scale.

Ashgarshirazi et al. [41] analyzed the data of 88 children out of 120 with abdominal pain-related functional gastrointestinal disorders except for abdominal migraine. The study included 30 patients with IBS. It was a randomized placebo-controlled trial intended to compare the effects of peppermint oil (34 patients), a symbiotic (*Bacillus coagulans* and fructooligosaccharide) (29 patients), and placebo (folic acid) (25 patients). Treatment was administered for one month. The primary outcomes were changes in severity, duration, and frequency of pain. Parents and patients reported the severity of pain based on a numerical rating scale from one to ten. The duration of pain was expressed in minutes per day. The number of episodes per week was recorded to evaluate the frequency of pain. There were significant changes in the group who received peppermint oil, in all the measured parameters, compared with the placebo group (*p* < 0.0001). Pain duration and severity were significantly decreased in the group that received peppermint oil compared with the group that received the symbiotic. There were no adverse reactions described [41]. 

The effects of three different doses (180, 360, and 540 mg) of peppermint oil on microbiome composition were investigated by Thapa et al. [42] in a study on 30 children aged 7 to 12 years with functional abdominal pain disorders, including IBS. Each group of 10 participants took the respective dose for a week. Stool samples were collected before and after this period, and microbiome composition was analyzed. There were no significant differences between baseline microbiome composition and after the administration of peppermint oil, regardless of the doses used. However, there were differences in the abundance of Collinsella, and the Firmicutes/Bacteroidetes ratio was lower in children who received 540 mg than in those who received 180 and 360 mg of peppermint oil. The authors concluded that further studies are needed to detect possible changes in gut microbiome as it seems that, from this study, the clinical benefit of peppermint oil may not be mediated through this path [42].

In the same group of patients, Shulman et al. [43] evaluated the pharmacokinetics of peppermint oil, trying to correlate the antispasmodic effects of three different doses of peppermint oil in children with functional gastrointestinal disorders, including IBS. Intestinal transit times were determined by wireless motility capsule before and after the treatment with 180, 360, and 540 mg of peppermint, respectively. Over 24 h, the pharmacokinetics of menthol was determined by blood sampling. Neither small bowel nor colonic transit seemed to be affected by peppermint oil. The whole gut mean contraction amplitude decreased after the administration of peppermint oil compared to baseline values. Higher doses may be needed to determine the threshold dose for menthol effects on the gut [43].

Ginger is extracted from the root of *Zingiber officinale Roscoe*. It is known for its antiemetic effects and is used to treat nausea and vomiting in pregnancy, gastroenteritis, dyspepsia, chemotherapy etc. [44]. Ginger also has anti-inflammatory, antimicrobial, antioxidant, and hepatoprotective effects [45]. Its action on the lower gastrointestinal tract is disputed. Ghayur et al. showed that ginger acts through cholinergic and calcium antagonist mechanisms leading to a combination of spasmolytic and spasmogenic effects [45]. On the other hand, one of the components in ginger, 6-gingerol, inhibits the production of prostaglandin E2 and, thus, the severity of pain [44]. Ginger has not shown beneficial effects in adult patients with IBS compared to a placebo [46,47]. Ciciora et al. [8] conducted a survey about the use of complementary and alternative therapies in 100 children with functional abdominal pain disorders, 47 of them with IBS (mean age 13.5 ± 3.5 years). As reported in the study, five children (11%) used ginger in their therapy [8]. There are no randomized controlled trials or other studies on children with IBS.

Turmeric derives from the roots of *Curcuma longa*. Curcumin, a yellowish pigment, is a purified extract of turmeric. Antioxidant, anti-inflammatory (by suppressing cytokine production), hepatoprotective, antiallergic, and other effects have been attributed to curcumin and turmeric [48]. The anti-inflammatory and antioxidant effects also extend to the gastrointestinal system, which also influences permeability, microbiota, and infections (bacterial, parasitic, and fungal) [49]. However, there are concerns regarding curcumin’s low bioavailability and absorption, and its fast elimination. The pharmacokinetics of curcumin seems to be improved in combination with other herbal extracts [48]. In adults with IBS, curcumin seems to have efficacy on gastrointestinal symptoms, both in a multi-herbal combination [50,51] and as a sole component [49,52]. In the study mentioned above by Ciciora et al., 2/47 (4%) children with IBS reported using curcumin or turmeric in their treatment [8]. Studies in children with IBS regarding curcumin’s efficacy and side effects are lacking.

Fennel (*Foeniculum vulgare*) is an aromatic herb, and oil is extracted from its seeds. The major component of the oil extract is anethole, which is similar in chemical structure to dopamine and has a relaxing effect on the intestinal smooth muscles, thus reducing abdominal pain in patients with IBS [53]. A study on adult patients with IBS showed promising results of fennel oil in combination with curcumin for the symptoms and quality of life [52], but there is no evidence in children with IBS.

Cannabis (*Cannabis sativa*) contains two major cannabinoids: delta-9-tetrahydrocannabinol and cannabidiol [54]. They might play an important role in intestinal physiology through the endocannabinoid system and its different receptors [55]. This herb might help alleviate pain in patients with IBS, but no statistically significant difference was found in pain scores in adult patients with IBS who were treated with cannabidiol instead of those who received a placebo [56]. No studies on children with IBS were found regarding treatment with extracts from cannabis.

The artichoke (*Cynara scolymus*) leaf extract reduces cholesterol levels and has a choleretic, spasmolytic, and antiemetic effect [57]. Studies in adult patients with IBS report reduced symptoms [58,59], but results in children with IBS have not been reported yet.

Senna is the name of a leaf or fruit of *Senna alexandrina* or *Cassia senna* (*C. acutifolia*) and of *Cassia augustifolia* (Tinnevelly senna, Indian senna, Mecca senna) [60], used as anthranoid glycosides laxatives. Senna contains sennosides, β glucosides of sennidin that pass unabsorbed through the stomach and the small intestine, reaching the colon where they are metabolized by the β glucosidases of the colonic bacteria [61]. Senna increases intestinal motility through its action on the enteric nerves of the intestinal smooth muscles [62]. Senna has been used as a laxative in patients with chronic constipation [63] or for bowel preparation before colonoscopy [64], including in children [65]. Abdominal cramps, nausea, vomiting, bloating, flatulence, diarrhea, perineal blisters, and perineal rash are adverse effects described in some patients, including children [62]. No studies report on the benefits and the adverse effects in children with IBS.

Aloe vera [11,66,67] was studied in adult patients with IBS, but symptoms improvement was only reported on short-term [67], and there are concerns about side effects [66]. The potential pharmacological benefits of rhubarb peony decoction in IBS are being studied [68]. There are no studies in children with IBS concerning these herbs.

Recent guidelines for managing IBS, developed with the participation of pediatrics societies, recommend soluble fibers in treating IBS, although the level of evidence is low [20]. Insoluble fibers are not recommended. For cannabinoid and endocannabinoid modulators, there is insufficient evidence supporting their use in treating IBS symptoms [20]. 

Our review focused on dietary fibers, herbal remedies, and spices that are used in IBS. We did not search for evidence for all types of dietary fibers. A paper on the use of dietary fibers in healthy children, and in pediatric gastrointestinal disorders in general, has recently been published [69]. We did not discuss the subject of prebiotics’ effectiveness in IBS, although some of the fibers that we mentioned have this role.

We found very few good quality studies related to alternative therapies in children with IBS. Psyllium is efficient in children with IBS in the short term. However, a more prolonged therapy and follow-up is needed to evaluate both its effectiveness and relapse rate in adequately designed randomized controlled trials. 

Many studies included children with abdominal-pain-related functional gastrointestinal disorders in general and were not specifically focusing on IBS. Studies were found to have a small number of participants, a short follow-up period, and some used symptom scales validated for adults, but not children, with IBS. For most herbal remedies, good quality studies in children are lacking.

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
