# Peer review of "The Use of Fibers, Herbal Medicines and Spices in Children with Irritable Bowel Syndrome: A Narrative Review"

_nutrients, 2023, doi:10.3390/nu15204351_

Round 1

Reviewer 1 Report

The recession of manuscript No. 2612286: „ The use of herbal medicines and spices in children with irritable bowel syndrome: A narrative review, written by Daniela Pop, Radu Samuel Pop, Dorin Farcău, “which will be published in Nutrients.

            The structure of the manuscript has the commonly required criteria. The topic of the presented work is very actual. The etiology of irritable bowel syndrome (IBS) is still unknown. A wide range of mechanisms implicate the microbiota-gut-brain axis, which might be involved in the pathophysiology of IBS.

            In the present study, the authors investigated current evidence for the benefits and side effects of herbal remedies and spices in pediatric patients with IBS. It is known that dietary fibers promote digestive health by influencing stool consistency and frequency of defecation, the microbiota, and metabolism. Herbal remedies used in traditional medicine have pharmacologically different components, and their synergistic effect could target multiple mechanisms and thus be of benefit in such a complex disease as inflammatory bowel disease (IBD).

The authors found that recent guidelines for managing IBS, developed with the participation of pediatric societies as well, recommend soluble fibers in treating IBS. are not For cannabinoid and endocannabinoid modulators, there is insufficient evidence to be used in treating IBS symptoms. The authors found few good-quality studies on alternative therapies in children with IBS. Psyllium is efficient in children with IBS in the short term. Still, prolonged treatment and follow-up are needed to evaluate its effectiveness and relapse rate in adequately designed randomized controlled trials. The results are documented in the table that presents the review of the obtained data.  

            The citations are well-chosen and relevant, and their format respects usual standards. The conclusion summarizes the author´s results.

            In summary, the manuscript can be published. 

Author Response

Distinguished reviewer,

Thank you for taking the time to evaluate our work. We are glad you considered our work valuable and worthy of sharing with other medical professionals. Our purpose was to increase awareness of this topic in the Pediatric community as more and more parents ask for alternative therapies, and it is essential to advise them based on current evidence.

Thank you!

Kind regards,

Daniela Pop

Reviewer 2 Report

This review is indeed needed to as there are not many comprehensive reviews on the topic; however, there are major revisions needed as follows:

1)     The definition of dietary fibre is focused on polysaccharides which is not up to date with most recent definition set by Codex that includes oligosaccharides.

2)     Not sure why Senna leaves, polyols, fig syrup, probiotics and prebiotics are not separately included in the reviews as alternative medicines as the focus here does not seem to be solely on herbal supplements. Psyllium husk, glucomannan, and guar gum are not herbal products anyway while senna leave is.

3)     The use of probiotics at various pages is a spelling error and it should be replaced with “prebiotics”. Probiotics are live microbial cells and prebiotics are types of fibre.

Other than the points above, I see very unbiased view and evaluation of the current literature on those lists of herbal/alternative treatments which is something to be commended.

The use of “we” at various pages needs some consideration compare do the use of passive tense.

Author Response

Distinguished reviewer,

Thank you for taking the time to evaluate our work. We are glad you considered our work valuable and worthy of sharing with other medical professionals. You raised some important issues, and we addressed them as follows: (our answers are highlighted)

  • The definition of dietary fibre is focused on polysaccharides which is not up to date with most recent definition set by Codex that includes oligosaccharides.

Based on your remark, we corrected the definition of dietary fibers, including oligomers. (page 2, lines 71-72)

  • Not sure why Senna leaves, polyols, fig syrup, probiotics and prebiotics are not separately included in the reviews as alternative medicines as the focus here does not seem to be solely on herbal supplements. Psyllium husk, glucomannan, and guar gum are not herbal products anyway while senna leave is.

We included a fragment about Senna in the revised version of the manuscript (page 8, lines 296-306). We did not include it initially because it is mainly used in children with chronic constipation (and the studies we found included patients diagnosed based on the definition of this functional disorder). However, since IBS can also be manifested with constipation, it is worth mentioning its lack of evidence in IBS. Of course, many other herbal products were not mentioned in this paper, including fig syrup, and the main reason for this is the deficient evidence, even in adults or other diseases in children for these preparations. Polyols, prebiotics (although some of the mentioned products also act as prebiotics), and probiotics were debated or will be debated in other papers, and this is why we did not include them in this article.

We included in the title the word “fibers.” We explained in a sentence at the end of the paper that we only covered in this paper some fibers.  (page 9, lines 316-320)

  • The use of probiotics at various pages is a spelling error and it should be replaced with “prebiotics”. Probiotics are live microbial cells and prebiotics are types of fibre.

We apologize for this typing mistake. Thank you for the remark. We also used a computer program to help improve English quality, and “prebiotic” was changed to “probiotic.”

Comments on the Quality of English Language

The use of “we” at various pages needs some consideration compare do the use of passive tense.

We made the corrections.

Thank you!

Kind regards,

Daniela Pop

Reviewer 3 Report

The scientific article presented to me for evaluation deals with the topic of supportive treatment of IBS in children. 

The work has the character of a review paper. 

The authors analyzed several scientific papers on the above topic.

The paper contains 1 table.

The paper cites 61 items of scientific bibliography.

In my opinion, the work qualifies for publication in a popular journal for parents. It has little value for professional scientists. I do not recommend acceptance of the work for publication in Nutrients.

Author Response

Distinguished reviewer,

Thank you for taking the time to evaluate our work. We value your point of view. The idea of this search started precisely from the parents of our patients who either seek our advice regarding alternative therapies in children with irritable bowel syndrome or have already started administering their children such products, many times without informing the Pediatrician or the Pediatric Gastroenterologist about this course of action. However, it is essential to increase awareness about this topic, specifically regarding children with irritable bowel syndrome in the medical community, so that the medical professional can provide evidence-based advice.

Thank you!

Kind regards,

Daniela Pop

Round 2

Reviewer 2 Report

The authors have not made the necessarily changes and/or replied ot the questions raised and no futhur changes required in my view.

Reviewer 3 Report

I propose to accept the wI propose to accept the work for publication in its present form.ork for publication in its present form.